# The Light-Controlled Release of 2-fluoro-l-fucose, an Inhibitor of the Root Cell Elongation, from a nitrobenzyl-caged Derivative

**DOI:** 10.3390/ijms24032533

**Published:** 2023-01-28

**Authors:** Mathieu Carlier, Thomas Poisson, Jean-Claude Mollet, Patrice Lerouge, Cyrille Sabot, Arnaud Lehner

**Affiliations:** 1Université de Rouen Normandie, GLYCOMEV UR 4358, SFR Normandie Végétal FED 4277, Innovation Chimie Carnot, IRIB, F-76000 Rouen, France; 2Université de Rouen Normandie, INSA Rouen Normandie, CNRS, COBRA UMR 6014, Innovation Chimie Carnot, F-76000 Rouen, France; 3Institut Universitaire de France, Rue Descartes, F-75231 Paris, France

**Keywords:** cell wall biosynthesis, glycosyltransferase inhibitor, glycan engineering, plant cell elongation, 2-fluoro fucose, caged inhibitor, photoirradiation

## Abstract

Glycan metabolic engineering is a powerful tool for studying the glycosylation in living plant cells. The use of modified monosaccharides such as deoxy or fluorine-containing glycosides has been reported as a powerful pharmacological approach for studying the carbohydrate metabolism. 1,3,4-tri-*O*-acetyl-2-fluoro-l-fucose (2F-Fuc) is a potent inhibitor of the plant cell elongation. After feeding plant seedlings with 2F-Fuc, this monosaccharide derivative is deacetylated and converted by the endogenous metabolic machinery into the corresponding nucleotide-sugar, which then efficiently inhibits Golgi-localized fucosyltransferases. Among plant cell wall polymers, defects in the fucosylation of the pectic rhamnogalacturonan-II cause a decrease in RG-II dimerization, which in turn induce the arrest of the cell elongation. In order to perform the inhibition of the cell elongation process in a spatio-temporal manner, we synthesized a caged 3,4-di-*O*-acetyl-1-hydroxy-2-fluoro-l-fucose (1-OH-2F-Fuc) derivative carrying a photolabile *ortho*-nitrobenzyl alcohol function at the anomeric position: 3,4-di-*O*-acetyl-1-ortho-nitrobenzyl-2-fluoro-l-fucose (2F-Fuc-NB). The photorelease of the trapped 1-OH-2F-Fuc was performed under a 365 nm LED illumination. We demonstrated that the in planta elimination by photoexcitation of the photolabile group releases free 2F-Fuc in plant cells, which in turn inhibits in a dose-dependent manner and, reversibly, the root cell elongation.

## 1. Introduction

Glycan metabolic engineering takes advantage of the cellular metabolism and associated enzymes to modify the structures of glycans in living organisms [1]. One of the most widely used glycan metabolic engineering strategies is the in vivo labeling of cell surface glycomolecules. In this method, a chemical reporter is introduced in surface glycomolecules using the endogenous metabolic machinery of the cell. This reporter is then covalently coupled through an efficient click-mediated reaction to a fluorescent probe added exogenously. These labeling experiments were performed by feeding cells with monosaccharides containing a bio-orthogonal chemical reporter, such as an azide or alkyne group [2]. The azide and alkyne chemical reporters can be attached either to the glycomolecule or to the probe, and the chemical coupling is catalyzed by copper (I) ions in biological conditions [3]. In plants, the labeling of cell wall polysaccharides was also performed by feeding plant cells with per-acetylated alkyne or azido-sugars and then chemical ligation to a fluorescent azide- or alkyne-containing fluorescent probe allowing efficient labeling of the primary cell wall of plant roots (Figure 1A) [4,5,6,7,8,9].

In addition to the cell surface labeling, the use of glycosyltransferase inhibitors is another aspect of the glycan metabolic engineering (Figure 1B). Glycosyltransferases use activated nucleotide-sugars (ADP-, UDP-, GDP- or CMP-monosaccharide) as a donor substrate in glycosylation processes. The transfer from the nucleotide-sugars of the monosaccharide on substrates occurs through the formation of an intermediate oxocarbenium ion within the catalytic site of glycosyltransferases. 2-fluorinated nucleotide-sugars are known to be potent glycosyltransferase inhibitors. Indeed, the presence of the fluorine atom at position 2 of monosaccharides induces an electronic effect that destabilizes the formation of the intermediate oxocarbenium ion that leads to a complex inhibiting the enzyme catalytic site. Plant cell walls are composed of three classes of structural polysaccharides: cellulose, hemicelluloses and pectins, which form a complex extracellular matrix. Reverse genetic experiments targeting Golgi-localized glycosyltransferases have provided numerous data concerning the function of cell wall polysaccharides in plant development. Recently, the use of fluorinated sugars was reported to be an alternative pharmacological approach for the inhibition of glycosyltransferases [10,11]. As an illustration, 1,3,4-tri-*O*-acetyl-2-fluoro-l-fucose (2F-Fuc) was demonstrated to inhibit the fucosylation of cell wall polysaccharides and the protein *N*-glycosylation at micromolar concentrations in plant root seedlings. 2F-Fuc is able to inhibit all plant fucosyltransferases [11]. In these experiments, plant seedlings were fed with 1,3,4-tri-*O*-acetyl-2-fluoro-l-fucose. As for the chemical reporter strategy (Figure 1A), after passive diffusion through the plasma membrane, this 2F-Fuc derivative is deacetylated. 2F-Fuc is then converted into GDP-2F-Fuc by the cytosolic endogenous metabolic machinery. Finally, this fluorinated nucleotide-sugar derivative is imported in the Golgi apparatus where it is able to inhibit Golgi-localized fucosyltransferases (Figure 1B). Cytosolic accumulation of GDP-2F-Fuc also triggers the shutdown of the de novo pathway that synthesizes endogenous GDP-Fuc nucleotide-sugar [12] (Figure 1B). The inhibition of the fucosylation of protein *N*-glycans and of cell wall xyloglucan did not induce developmental phenotypes in plants [11]. However, 2F-Fuc also affected the fucosylation of the pectic rhamnogalacturonan-II (RG-II), a highly complex cell wall polymer containing a fucose residue in its B side-chain [13]. Treatment of *Arabidopsis thaliana* seedlings with 2F-Fuc inhibitor was shown to induce a decrease in RG-II biosynthesis and its dimerization, which in turn resulted in the alteration of the root growth [11]. Closer investigation of the growth phenotype induced by 2F-Fuc demonstrated that cell division was not affected. In contrast, the 2F-Fuc fully suppressed the elongation of root cells. Therefore, 2F-Fuc is considered a specific and efficient inhibitor of plant root cell elongation (Figure 1B). 

Caged molecules were developed for the study of specific physiological processes. Caged compounds are biologically inert whilst photosensitive molecules that are able to release the trapped compounds at high speed into biologically active molecules by photoirradiation. Therefore, in vivo photolysis of the caged compound by illumination at specific wavelengths enables the control of biological processes with a high spatio-temporal specificity [14]. Various caged molecules, including sugars, have been chemically synthesized, and the conditions for their efficient in vivo photolysis have been investigated [15]. In contrast, the use of caged molecules for the study of plant physiological processes has received little attention to date [16,17,18]. In fact, the light of the growth chamber can interfere and cause an uncontrolled photolysis of the compound, thus limiting experiments to dark growing cells. In this article, we investigated the synthesis and the in planta use of an *ortho*-nitrobenzyl alcohol-containing caged derivative of 3,4-di-*O*-acetyl-1-hydroxy-2-fluoro-l-fucose (1-OH-2F-Fuc). We demonstrated that the caged compound is stable throughout the growth of the plant in our light culture conditions and that the in planta elimination by specific photoexcitation of the photolabile group releases free 1-OH-2F-Fuc in the plant cell, which in turn inhibits the root cell elongation (Figure 1C).

## 2. Results

### 2.1. Chemical Synthesis and In Vitro Photolysis of the Caged 3,4-Di-O-acetyl-1-ortho-nitrobenzyl-2-fluoro-l-fucose (2F-Fuc-NB)

We chose to protect the anomeric hydroxyl position of 1-OH-2F-Fuc with *ortho*-nitrobenzyl alcohol derivatives to obtain photosensitive caged-2F-Fuc derivatives that cannot be metabolized by the plant cell into a biologically active nucleotide-sugar. The photoexcitation of *ortho*-nitrobenzyl alcohol derivatives induces the formation of *aci*-nitro derivatives that give rise to a bicyclic intermediate. The rearrangement of this intermediate then releases the anomeric hydroxyl group of the 1-OH-2F-Fuc and an *ortho*-nitrosobenzaldehyde [19] (Figure 2A). Two *ortho*-nitrobenzyl alcohol derivatives were considered: *ortho*-nitrobenzyl alcohol (NB) (λ_abs._~230–380 nm) and 4,5-dimethoxy-*ortho*-nitrobenzyl alcohol (λ_abs._~260–420 nm). However, *Arabidopsis* plants are grown in a growth chamber under RGB LED-type lighting ranging from λ = 400 to 800 nm, a wavelength range which overlaps with the λ_abs._ range of 4,5-dimethoxy-*ortho*-nitrobenzyl alcohol. Thus, we selected *ortho*-nitrobenzyl alcohol for the protection of 1-OH-2F-Fuc with a photolysis of the cage performed at 365 nm LED lighting.

The synthesis of caged-1-OH-2F-Fuc first required the synthesis of free fluorinated fucose. The synthesis began by the l-fucose per-acetylation followed by an anomeric carbon bromination. The resulting bromo glycoside was then converted into the per-acetylated l-fucal by reductive Fisher–Zach removal [20]. This glycal was used in an oxo-fluorination reaction with Selectfluor^®^ to obtain a mixture of the α and β anomers of the 3,4-di-*O*-acetyl-1-hydroxy-2-fluoro-l-fucose, which was then acetylated to obtain a mixture of α/β anomers 1,3,4-tri-*O*-acetyl-2-fluoro-l-fucose (its C-2 epimer, 6-desoxy-2-fluoro-l-talose, only accounted for less than 2%) [21]. This glycoside underwent an anomeric carbon iodination reaction followed by a Koenigs–Knorr glycosylation reaction with the *ortho*-nitrobenzyl alcohol in the presence of silver perchlorate (Figure 2B) to give a mixture of α/β anomers of 3,4-di-*O*-acetyl-1-*ortho*-nitrobenzyl-2-fluoro-l-fucose (2F-Fuc-NB) [22] (Appendix A).

1-OH-2F-Fuc was synthetized to provide an authentic NMR reference of the product photoreleased from the cage. Its synthesis was achieved by a mono-deacetylation of the anomeric acetate of 2F-Fuc using ammonia in methanol [23].

Stability and in vitro photolysis of 2F-Fuc-NB were then investigated. 2F-Fuc-NB was found to be stable in the growth chamber illumination conditions over a 7-day period. Its photoirradiation in vitro assay was then performed using a 365 nm light irradiation, and formation of the 1-OH-2F-Fuc was monitored by proton-uncoupled ^19^F and ^1^H NMR (Appendix A). After 1 min of irradiation, the caged molecule was fully converted into an α/β mixture of the 1-OH-2F-Fuc (Appendix A). After 7 days in the dark, the samples were further analyzed and showed no noticeable degradation (Appendix A). The photoirradiation at 365 nm of a mixture of 2F-Fuc-NB and 1-OH-2F-Fuc shows the disappearance of 2F-Fuc-NB signals and the increase in signals corresponding to 1-OH-2F-Fuc (Appendix A).

### 2.2. In Planta Assays

In order to test the possibility of inhibiting root elongation after photorelease of 1-OH-2F-Fuc from its caged form, we investigated the effects of 2F-Fuc on root growth under our culture conditions. In fact, these conditions are different from those presented in Dumont et al. 2015 [11], which highlighted the inhibition of cell elongation in the root elongation zone in the presence of 2F-Fuc. We confirmed that the inhibitory effects are reversible when seedlings are treated with 2F-Fuc for 3 days and then transferred to a control medium for an additional 4 days (Figure 3). The dose-dependent inhibition of root cell elongation is statistically significant in the medium supplemented with 1 µM of 2F-Fuc, with a mean reduction of the root length of 32% compared to control conditions (Figure 3A). The effects are dose-dependent and reached a plateau for concentrations above 2.5 µM and up to 10 µM, with a reduction of the root length of 75% and 84%, respectively (Figure 3A). For a higher concentration, i.e., 16 µM, the reduction of the root length reached 87% (Figure 3B). Interestingly, the effect of the treatment is reversible, meaning that the fluorinated sugar is not lethal for the meristem. In Figure 3B, the seeds germinated on medium supplemented with increasing concentration of 2F-Fuc and the seedlings grew for 7 days on the medium. This resulted in an important decrease in the root length (Figure 3B–G) and an almost complete absence of the elongation zone in treated roots, as shown in Figure 3O, compared to the control condition (Figure 3N), in which the elongation zone is clearly visible. As shown in Dumont et al. 2015 [11], the meristematic cells are not affected by 2F-Fuc; thus, we could speculate that the effect of 2F-Fuc is reversible if seedlings are transferred on the control medium. We confirmed in Figure 3B (right panel) and in Figure 3H–L,P that the transfer of the seedlings on a control medium allows the recovery of a normal growth phenotype after 4 days, with the reappearance of the elongation zone of the root, as is clearly visible in Figure 3P. These experiments confirmed that the root cell inhibition by 2F-Fuc is similar to what was found in the literature under our culture conditions and allowed the use of this sugar analog as a positive control in the following experiments (Appendix A). 

Then, we investigated the physiological effects induced by 2F-Fuc-NB. Figure 4 shows the dose-dependent effect of 2F-Fuc-NB on *Arabidopsis* root growth after 7 days in the growth chamber without (Figure 4A) or with the photorelease of the 1-OH-2F-Fuc (Figure 4B) after 3 days of growth. When seedlings were germinating on a medium supplemented with 2F-Fuc-NB without photolysis, the mean root length was statistically longer for concentrations ranging from 1 to 8 µM compared to the control condition (Figure 4A). Nevertheless, the difference between treated and control plants never exceeded 20%. In contrast, when Petri dishes were exposed to 365 nm LED lighting for 10 min after 3 days of growth, the mean root length of the root drastically decreased in a dose-dependent manner for concentrations ranging from 1 to 10 µM (Figure 4B). The reduction of the mean root length was comparable to the one observed with 2F-Fuc (Figure 3A), i.e., 75 to 80% of reduction for concentrations above 5 µM, meaning that the 1-OH-2F-Fuc was successfully freed in the medium after photolysis.

To ascertain if 2F-Fuc-NB was efficiently incorporated into the root cells before photolysis, we sowed *Arabidopsis* seeds on the control medium or in the presence of 5 µM of 2F-Fuc or 5 µM of 2F-Fuc-NB. Seeds were germinated, and seedlings grew for 3 days in the growth chamber before photolysis (Figure 5A, right panel) or were transferred after 3 days of culture to the control medium before photolysis (Figure 5B, right panel). These results clearly showed that the seedlings treated with 2F-Fuc or 2F-Fuc-NB and then irradiated for photorelease 1-OH-2F-Fuc displayed an alteration of the root growth (Figure 5A) associated with an absence of the elongation zone in the case of 2F-Fuc and 2F-Fuc-NB (Figure 5H). In contrast, the control seedlings irradiated or not did not show any observable or measurable root phenotype (Figure 5A,C,D). Finally, when the seedlings were incubated in the presence of 5 µM of 2F-Fuc-NB without photoirradiation, no effect was observed (Figure 5A,G).

Conversely, when the seedlings were grown on a medium containing 5 µM of 2F-Fuc-NB and transferred to the control medium before being irradiated for photolysis, the effects on root growth inhibition were visible, with a decrease in the root length of 20% 4 days after the photolysis (Figure 5B). The phenotype is then quite marked, with an absence of the elongation zone (Figure 5J), unlike the irradiated control seedlings transferred on a new medium, in which the root develops normally with a well-delimited elongation zone (Figure 5F). These experiments confirmed that 2F-Fuc-NB is efficiently incorporated in the roots and that the photoexcitation is effective and releases 1-OH-2F-Fuc that in turn inhibited the root cell elongation.

## 3. Discussion

Since pioneering work published by Kaplan et al. (1978) [24] on ATP using an *ortho*-nitrobenzyl caging group and its successful release in blood cells, few research has been conducted on plant cells. The role of calcium signaling during cell growth engaged the attention of scientists working on guard cells and pollen tube growth. Calcium and inositol (1,4,5)-triphosphate were photoreleased from caged derivatives to provide evidence of their roles during cell signaling and cell growth [25,26,27,28,29]. Works have also been carried out using caged plant growth regulators [30] such as abscisic acid [31] and, more recently, with 1-naphthaleneacetic acid, a synthetic auxin [16]. Nevertheless, to date, no data have been reported on caged inhibitors of enzymes involved in cell wall biosynthesis and therefore in cell elongation.

In order to perform the spatio-temporal inhibition of the plant cell elongation, we synthesized a 1-OH-2F-Fuc derivative protected with an *ortho*-nitrobenzyl alcohol function on its anomeric position. In plants, 2F-Fuc was previously shown to be metabolically incorporated in the polysaccharide fucosylation pathway and to induce defects in root elongation [10,11]. The caged 1-OH-2F-Fuc derivative was found to be stable under light conditions used for *Arabidopsis* growth. In fact, the light of the growth chamber would interfere and cause the photolysis of the caged compound, thus limiting experiments to dark growing cells [25,26,27,28,29]. In contrast, the photoirradiation with a 365 nm LED illumination provoked a photo-induced cleavage mechanism, resulting in the fast in vitro release of 1-OH-2F-Fuc. Physiological effects of the caged and noncaged 2F-Fuc on *Arabidopsis* seedlings were then investigated. In contrast to the 2F-Fuc, which induced, in a reversible manner, the root growth inhibition at micromolar concentrations (IC_50_ = 1.1 µM) (as reported in literature [10]), 2F-Fuc-NB did not induce significant effects on the roots when used at the same concentrations and without irradiation at 365 nm. This indicated that the caged inhibitor is not toxic for *Arabidopsis* plants, and that, as expected, the photorelease of 1-OH-2F-Fuc did not occur in the growth chamber illumination conditions over a 7-day period. This observation also demonstrated the stability of the photolabile cage toward plant metabolic processes (notably plant hydrolases) and that this derivative efficiently prevents its in planta activation into GDP-2F-Fuc. In contrast, when plants were exposed to a 365 nm LED lighting for 10 min after 3 days of growth, the root length of *Arabidopsis* roots drastically decreased in a dose-dependent manner (IC_50_ = 2.9 µM) with intensities that were comparable to the ones observed with 2F-Fuc. 

To investigate whether the photorelease of the 2F-Fuc inhibitor occurred after incorporation into the root cells, *Arabidopsis* seedlings were transferred after 3 days of culture in the presence of the 2F-Fuc-NB to the control medium before photoirradiation. We clearly demonstrated that the seedlings treated with 2F-Fuc-NB and then irradiated at 365 nm displayed an alteration of the root growth at a wavelength that did not induce any detectable physiological effects in control experiments. As for plants treated with 2F-Fuc, this growth inhibition is due to the absence of the elongation zone. Taken together, these results indicated that the photorelease at 365 nm of 1-OH-2F-Fuc from the caged inhibitor occurred after its incorporation into the root cells, which in turn induced the inhibition of the root cell elongation. 

We concluded that the caged 1-OH-2F-Fuc, which was chemically synthesized in this study, is not toxic for *Arabidopsis* plants and that the photorelease of this cell elongation inhibitor occurred in planta by irradiation at 365 nm and in turn affected the root growth. Consequently, 2F-Fuc-NB is a suitable caged drug for a spatio-temporal inhibition of cell elongation by confocal microscopy photoirradiation. It therefore highlighted the potential of photoreleased glycosyltransferase inhibitors for studying the function of cell wall polysaccharides during cell growth. The inhibition of other glycosyltransferase families, such as galactosyltransferase and arabinosyltransferase, may be considered in the near future on root cells and on other cell models. In fact, preliminary results have been obtained on pollen tubes and on the hypocotyl of *Arabidopsis thaliana*. In addition, the successful photo-release of phytohormones in root of mungo beans (*Vigna radiata*) [16] indicate that the use of photo-released glycosyltransferase inhibitors could be developed on various plant species, especially on plant of agro-economic interest. 

## 4. Materials and Methods

### 4.1. General Considerations

All chemicals were purchased from Merck (Fontenay-sous-Bois, France), Sigma-Aldrich (Saint Quentin Fallavier, France), Alfa Aesar (Heysham, Lancashire, UK), Acros Organics (Gell, Belgium). Solvents were purchased in reagent grade or HPLC grade. Dichloromethane and acetonitrile were obtained from an MB SPS-800 apparatus (MBRAUN, Mérignac, France). Chloroform and dichloromethane were stabilized on amylene. For nonaqueous reactions, flasks were dried by heating with a heat gun under vacuum. All reactions were monitored by thin-layer chromatography (TLC). TLC was carried out on DC Kieselgel 60 F-254 aluminum sheets (Merck). Visualization of spots was performed under a UV lamp at λ = 254 nm and/or staining with a vanillin solution in ethanol containing 2% H_2_SO_4_, developed by heating. Flash column chromatography purifications were performed manually on silica gel (40–63 µm) or C-18 silica gel under pressurized airflow.

^1^H, ^19^F and ^13^C NMR spectra were recorded on a 300 MHz Bruker FT-NMR piece of equipment operating at ambient probe temperature (Bruker, Wissembourg, France). The solvent resonance was used as the internal standard for ^1^H-NMR (CDCl_3_ at 7.26 ppm) and ^13^C-NMR (CDCl_3_ at 77.0 ppm). Chemical shifts (δ) were quoted in parts per million (ppm). Coupling constants (J) were quoted in Hertz (Hz). The following abbreviations were used to give the multiplicity of the NMR signals: s: singlet, d: doublet, dd: doublet of doublet, ddd: doublet of doublet of doublet, dddd: doublet of doublet of doublet of doublet. High-resolution mass spectrometry (HRMS) was performed with a Waters Micromass LCT Premier XE^®^ equipped with an orthogonal acceleration time-of-flight (oaTOF) and an electrospray source in positive or negative mode (Waters, Guyancourt, France). UV-Visible spectroscopy was performed on an Agilent Cary 60 UV-Vis^®^ spectrophotometer at room temperature (Agilent, Les Ulis, France).

The photoirradiation in vitro and in planta experiment was performed with an EvoluChem^TM^ LED 365PF (Wavelength 365 nm, LED/CREE, 9 mW/cm² (HepatoCHEM, Beverly, CA, USA)). The distance between LED and sample was fixed at 14 cm.

### 4.2. Chemical Synthesis of 3,4-Di-O-acetyl-1-ortho-nitrobenzyl-2-fluoro-l-fucose (2F-Fuc-NB)

Synthesis of 3,4-di-*O*-acetyl-l-fucal.

Perchloric acid (152 mg, 1.52 mmol, 0.1 eq.) was added slowly to a solution of l-Fucose (2.5 g, 15.23 mmol, 1.0 eq.) in dry anhydride acetic (8.6 mL, 91.37 mmol, 6.0 eq.) at 0 °C. The reaction mixture was stirred at room temperature under an argon atmosphere until acetylation was completed (followed by TLC). Then, at 0 °C, the reaction mixture was diluted by chloroform (10 mL) and phosphorous tribromide (0.9 mL, 2.5 g, 9.14 mmol, 0.6 eq.), and water (0.55 mL, 0.55 g, 30.46 mmol, 2.0 eq.) was slowly added. The reaction mixture was stirred at room temperature under an argon atmosphere. After bromination (followed by TLC), the reaction mixture was diluted by acetone (30 mL). A freshly prepared mixture of zinc dust (7.0 g, 106.6 mmol, 7.0 eq.), ammonium chloride (5.7 g, 106.6 mmol, 7.0 eq.) and copper (II) sulfate pentahydrate (250 mg, 1.5 mmol, 0.1 eq.) was slowly added at 0 °C. The reaction mixture was stirred at room temperature and air atmosphere overnight. The solid phase was removed by filtration on Celite^®^ plug, and the liquid phase was concentrated under vacuum. The crude material was diluted with ethyl acetate successively washed with HCl 1M, saturated aqueous NaHCO_3_ (×3) and brine. The organic layer was dried over anhydrous magnesium sulfate, filtered and concentrated under vacuum. The residue was purified by flash chromatography over silica gel (cyclohexane/Et_2_O, from 100:0 to 80:20) to give the expected product (2.8 g, 13.25 mmol, 87%) as a colorless oil. ^1^H NMR data of this compound were consistent with the literature data [31].

Synthesis of 1,3,4-tri-*O*-acetyl-2-fluoro-l-fucose (2F-Fuc).

To a solution of 3,4-di-*O*-acetyl-l-fucal (2.8 g, 13.25 mmol, 1.0 eq.) in a mixture of acetonitrile and water (0.5 M, 9:1) at room temperature, SelectFluor^®^ (9.3 g, 26.5 mmol, 2.0 eq.) was added. The reaction mixture was stirred at room temperature under an argon atmosphere overnight. After the total consumption of fucal (following by TLC), the reaction mixture was heated to 100 °C for 2 h. Then, the reaction mixture was concentrated under vacuum. The crude material was diluted with ethyl acetate and successively washed with HCl 1M, saturated aqueous NaHCO_3_ (×3) and brine. The organic layer was dried over anhydrous magnesium sulfate, filtered and concentrated under vacuum. Then, the crude material was dissolved in dry pyridine (10 mL) and dry anhydride acetic (3.7 mL, 39.75 mmol, 3.0 eq.) at 0 °C. The reaction mixture was stirred at room temperature under an argon atmosphere until acetylation was complete (followed by TLC). At 0 °C, the reaction mixture was stirred and slowly diluted by ethanol (10 mL). After 1 h at room temperature, the mixture was concentrated under vacuum. The crude material was diluted by ethyl acetate and successively washed with HCl 1M, saturated aqueous NaHCO_3_ (×3) and brine. The organic layer was dried over anhydrous magnesium sulfate, filtered and concentrated under vacuum. The residue was purified by flash chromatography over silica gel (cyclohexane/Et_2_O, from 100:0 to 70:0) to give a mixture of anomers (ratio 69:31 determined by ^1^H NMR analysis) of the expected product (2.1 g, 7.15 mmol, 54%) as a colorless oil. ^1^H NMR data of this compound were consistent with the literature data [32] and also indicated the presence in the final product of less than 2% of its C-2 epimer (1,3,4-tri-*O*-acetyl-2-fluoro-6-deoxy-l-talose).

Synthesis of 3,4-di-*O*-acetyl-1-ortho-nitrobenzyl-2-fluoro-l-fucose (2F-Fuc-NB).

Triethylsilane (149 mg, 205 µL, 1.28 mmol, 1.5 eq.) was added at 0 °C to a solution of 1,3,4-tri-*O*-acetyl-2-fluoro-l-fucose (250 mg, 0.86 mmol, 1.0 eq.) and iodine (326 mg, 1.28 mmol, 1.5 eq.) in dry chloroform (1.7 mL, 0.5 M). The reaction mixture was stirred at 40 °C under an argon atmosphere until TLC indicated complete conversion of the starting material. Then, saturated aqueous NaHCO_3_ (2 mL) and saturated aqueous Na_2_S_2_O_3_ (2 mL) were added at room temperature. After complete discoloration of the reaction mixture, chloroform was added (10 mL) and the organic layer was successively washed with HCl 1M, saturated aqueous NaHCO_3_ (×3) and brine. The organic layer was dried over anhydrous magnesium sulfate, filtered and concentrated under vacuum. Then, the crude material was dissolved in a mixture of dry dichloromethane and acetonitrile (1:1, 1.0 M). *Ortho*-nitrobenzyl alcohol (196 mg, 1.28 mmol, 1.5 eq.) and anhydrous silver perchlorate (266 mg, 1.28 mmol, 1.5 eq.) were added at 0 °C. The reaction mixture was stirred at room temperature under an argon atmosphere overnight. The solid phase was removed by filtration on Celite^®^ plug and the filtrate was concentrated under vacuum. The crude material was diluted with ethyl acetate and successively washed with HCl 1M, saturated aqueous NaHCO_3_ (×3) and brine. The organic layer was dried over anhydrous magnesium sulfate, filtered and concentrated under vacuum. The residue was purified by flash chromatography over silica gel (cyclohexane/Et_2_O, from 100:0 to 80:20) and by flash chromatography over C18-silica gel (H_2_O/CH_3_CN, from 100:0 to 50:50) to give a mixture of anomers (ratio 57:43 determined by ^1^H NMR analysis)) of the expected product (178 mg, 0.46 mmol, 54%) as a white powder after freeze-drying.

Chemical formula: C_17_H_20_FNO_8_. M = 385.34 g/mol; Rf: (SiO_2_, CH_2_Cl_2_/AcOEt, 9:1): 0.85.

HMRS (ESI^+^): [M+K]^+^: [C_17_H_20_FKNO_8_] calc *m*/*z*: 424.0810, found 424.0818.

NMR (Appendix A).

α anomer: ^1^H-NMR (300 MHz, CDCl_3_) δ: 8.12 (dd, *J* = 8.2, 1.2 Hz, 1H, Har), 7.92 (dd, *J* = 7.9, 1.2 Hz, 1H, Har), 7.68 (ddd, *J* = 7.9, 7.2, 1.2 Hz, 1H, Har), 7.46 (ddd, *J* = 8.2, 7.2, 1.2 Hz, 1H, Har), 5.29 (ddd, *J* = 3.4, 3.0, 1.1 Hz, 1H, H-4), 5.21 (d, *J* = 15.3 Hz, 1H, CHaH-ph), 5.12 (ddd, *J* = 9.7, 3.8, 3.4 Hz, 1H, H-3), 5.01 (d, *J* = 15.3 Hz, 1H, CHHb-ph), 4.70 (d, *J* = 7.6 Hz, 1H, H-1), 4.63 (ddd, *J* = 47.9, 9.7, 7.6 Hz, 1H, H-2), 3.87 (ddd, *J* = 12.8, 6.4, 1.1 Hz, 1H, H-5), 2.18 (s, 3H, CH_3_), 2.06 (s, 3H, CH_3_), 1.24 (d, *J* = 6.4 Hz, 3H, H-6). ^13^C-NMR (125 MHz, CDCl_3_) δ: 170.6, 170.3 (2 COCH_3_); 147.1 (C-ar), 134.2 (CH-ar); 128.7 (CH-ar); 128.4 (CH-ar); 125.0 (CH-ar); 100.4 (d, *J* = 22.4 Hz, C-1); 88.2 (d, *J* = 187.1 Hz, C-2); 71.6 (d, *J* = 15.9 Hz, C-3); 71.0 (d, *J* = 8.3 Hz, C-4); 69.6 (C-5); 66.8 (CH_2_-ph); 20.9, 20.7 (2 COCH_3_); 16.0 (C-6).^19^F (^1^H uncoupled)-NMR (282 MHz, CDCl_3_) δ: −207.08. 

β anomer: ^1^H-NMR (300 MHz, CDCl_3_) δ: 8.11 (dd, *J* = 8.2, 1.2 Hz, 1H, Har), 7.87 (dd, *J* = 7.9, 1.2 Hz, 1H, Har), 7.69 (ddd, *J* = 7.9, 7.2, 1.2 Hz, 1H, Har), 7.48 (ddd, *J* = 8.2, 7.2, 1.2 Hz, 1H, Har), 5.50 (ddd, *J* = 10.8, 10.3, 3.8 Hz, 1H, H-3), 5.36 (d, *J* = 15.3 Hz, 1H, CHaH-ph), 5.35 (ddd, *J* = 4.2, 3.8, 1.2 Hz, 1H, H-4), 5.19 (dd, *J* = 9.6, 3.8 Hz, 1H, H-1), 5.12 (d, *J* = 15.3 Hz, 1H, CHHb-ph), 4.84 (ddd, *J* = 50.0, 10.3, 3.8 Hz, 1H, H-2), 4.22 (ddd, *J* = 13.0, 6.6, 1.2 Hz, 1H, H-5), 2.17 (s, 3H, CH_3_), 2.06 (s, 3H, CH_3_), 1.15 (d, *J* = 6.6 Hz, 3H, H-6). ^13^C-NMR (125 MHz, CDCl_3_) δ: 170.6, 170.2 (2 COCH_3_); 147.1 (C-ar), 134.00 (CH-ar); 128.6 (CH-ar); 128.3 (CH-ar); 124.9 (CH-ar); 96.8 (d, *J* = 20.3 Hz, C-1); 85.6 (d, *J* = 190.9 Hz, C-2); 71.7 (d, *J* = 4.8 Hz, C-4); 68.8 (d, *J* = 18.6 Hz, C-3); 68.0 (CH_2_-ph); 65.3 (C-5); 20.8, 20.7 (2 COCH_3_); 15.8 (C-6). ^19^F (^1^H uncoupled)-NMR (282 MHz, CDCl_3_) δ: −208.63.

Synthesis of 3,4-di-*O*-acetyl-1-hydroxy-2-fluoro-l-fucose (1-OH-2F-Fuc).

NH_3_ in methanol (7 M, 370 µL, 2.59 mmol, 20.0 eq.) was added at 0 °C to a solution of 1,3,4-tri-*O*-acetyl-2-fluoro-l-fucose (50 mg, 0.13 mmol, 1.0 eq.) in dry THF (2.0 mL, 0.1 M). The reaction mixture was stirred at 0 °C under an argon atmosphere for 5 h. The reaction mixture was concentrated under vacuum. The residue was purified by flash chromatography over silica gel (DCM/Et_2_O, from 100:0 to 90:10) to give a mixture of anomers (ratio 57:43 determined by ^1^H NMR analysis) of the expected product (15 mg, 0.06 mmol, 46%) as a white solid. ^1^H, ^19^F and ^13^C NMR data of this compound were consistent with the literature data [33] and also indicated the presence in the final product of less than 2% of an unknown fluorinated impurity.

HMRS (ESI^−^): [M-H]^−^: [C_10_H_14_FO_6_] calc *m*/*z*: 249.0774, found 249.0775.

### 4.3. In Vitro Photorelease of 1-OH-2F-Fuc from 2F-Fuc-NB 

A solution of 2F-Fuc-NB and/or 1-OH-2F-Fuc in a mixture of D_2_O/DMSO-d_6_ (1:1, 600 µL, 2 mM) was added in an NMR tube, followed by the internal reference hexafluoroisopropanol (2 mM). This solution was analyzed by proton-uncoupled ^19^F and ^1^H NMR (t = 0), then irradiated at 365 nm and monitored over 10 min by ^19^F and ^1^H NMR analysis.

### 4.4. Plant Material and Growth Conditions

*Arabidopsis thaliana* Col-0 seeds were surface sterilized in 70% (*v*/*v*) ethanol for 5 min then in 30% (*v*/*v*) sodium hypochlorite for 2 min. Seeds were then washed 6 times in sterile water. Sterilized seeds were deposited on Murashige and Skoog (MS 2.2 g/L) mineral medium containing 0.8% plant agar. The plates were placed vertically at 4 °C for 48 h before being transferred in a growth chamber under a 16 h light/8 h dark cycle at 25 and 22 °C, respectively. Light equipment consisted of RGB LED light covering the visible spectra with the exception of UV light (Valoya^®^ RX500, model spectrum AP67, wavelength ratio: UV (<400 nm) 0%, Blue (400–500 nm) 12%, Green (500–600 nm) 16%, Red (600–700 nm) 57% and Far-red (700–800 nm) 16% (manufacturer’s data)).

For treatments with 2F-Fuc and 2F-Fuc-NB, molecules were dissolved in DMSO with a maximum DMSO concentration for the highest concentration of treatment (10 µM) corresponding to 0.25% of DMSO. Control conditions were performed in 0.25% DMSO. 2F-Fuc or 2F-Fuc-NB or DMSO (negative control) was added to the medium in a tube and homogenized by shaking before pouring the medium in the Petri dishes. Plants were grown for 7 days or gently transferred onto a treatment-free medium after 3 days and then placed for 4 more days in the growth chamber. Transfer of seedlings was achieved with plastic tweezers in order to avoid any damage.

### 4.5. In Planta Photorelease of 1-OH-2F-Fuc from 2F-Fuc-NB

After 3 days of growth on 2F-Fuc-NB supplemented medium, the Petri dish was exposed at room temperature (20 °C) to a 365 nm LED light for 10 min. Alternatively, after 3 days of growth on 2F-Fuc-NB supplemented medium, seedlings were transferred in a Petri dish containing control (treatment-free) medium. The Petri dish was then exposed to a 365 nm LED light for 10 min. The photolysis was always performed in the same manner, i.e., the bottom of the Petri dish faced the 365 nm LED. The Petri dish was then placed back in the growth chamber.

### 4.6. Image Acquisition

Seedlings were observed with an inverted microscope (Leica DMI 6000B), and images were acquired with the Leica DFC 450C camera. 

### 4.7. Root Length Measurements

After 7 days of growth, Petri dishes containing seedlings were scanned with a high-resolution scanner (HP Expression 11000 XL), and roots were measured using Image J software [34]. Sixty roots were measured for each replicate and each condition. Experiments were carried out in biological triplicates. Root length is expressed as % of the mean root length of the control condition for each experiment. Mean root length in the control condition ranged from 15.7 mm to 22.5 mm.

### 4.8. Statistical Analysis

Significant differences were determined by one-way ANOVA followed by Tukey’s HSD posthoc multiple comparison test. Data are marked by different letters when significantly different with respect to the control conditions (* *p*-value < 0.05, ** *p*-value < 0.01).

## Figures and Tables

**Figure 1 ijms-24-02533-f001:**
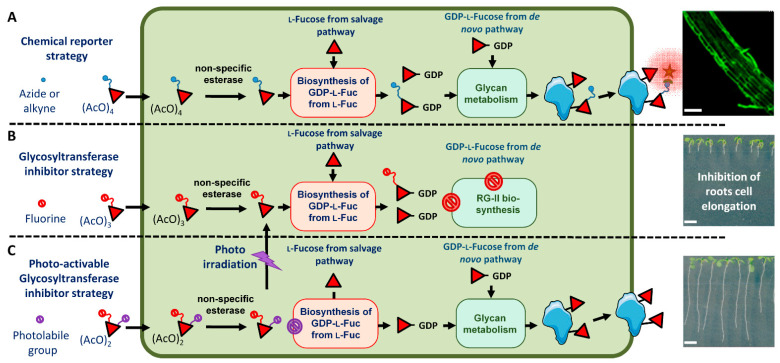
Glycan metabolic engineering in plant cells using fucose derivatives. (**A**) Feeding of seedlings with per-acetylated alkyne or azide fucose allows its incorporation in cell wall polysaccharides and subsequent coupling to fluorescent probes. The picture on the right shows metabolic labeling of *Arabidopsis* roots using CuAAC (copper(I)-catalyzed alkyne-azide cycloaddition) click-reaction with Fucose alkyne and Alexa 488 azide. (**B**) Feeding of seedlings with 2F-Fuc results in the inhibition of the root cell elongation and (**C**) feeding of seedlings with caged-2F-Fuc allows controlling the release of 2F-Fuc inhibitor by photoirradiation. Red diamond: l-fucose. Blue circle attached to diamond: chemical reporter (azide or alkyne function) for bio-orthogonal chemical conjugation with azide or alkyne-containing fluorescent probes (green star). Red circle attached to diamond: 2F-Fuc derivative. Purple circle attached to diamond: caged derivative. Scale bars: (**A**): 50 µM, (**B**,**C**): 5 mm.

**Figure 2 ijms-24-02533-f002:**
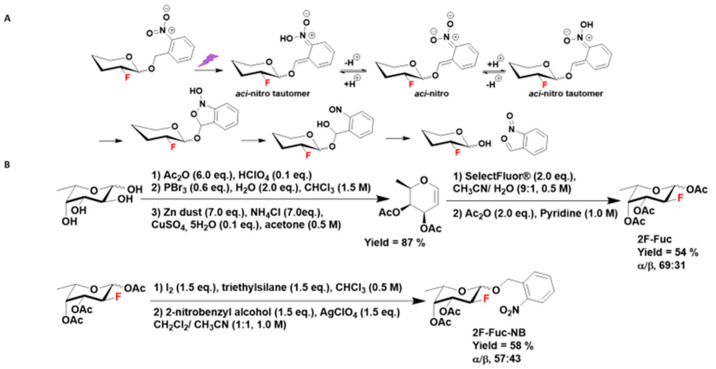
(**A**) Photoirradiation of 2-fluoro sugar caged by an *ortho*-nitrobenzyl alcohol on the anomeric carbon results in the release of free 2-fluoro monosaccharide and *ortho*-nitrosobenzaldehyde through *aci*-nitro intermediates. (**B**) Chemical synthesis of 3,4-di-*O*-acetyl-1-*ortho*-nitrobenzyl-2-fluoro-l-fucose (2F-Fuc-NB).

**Figure 3 ijms-24-02533-f003:**
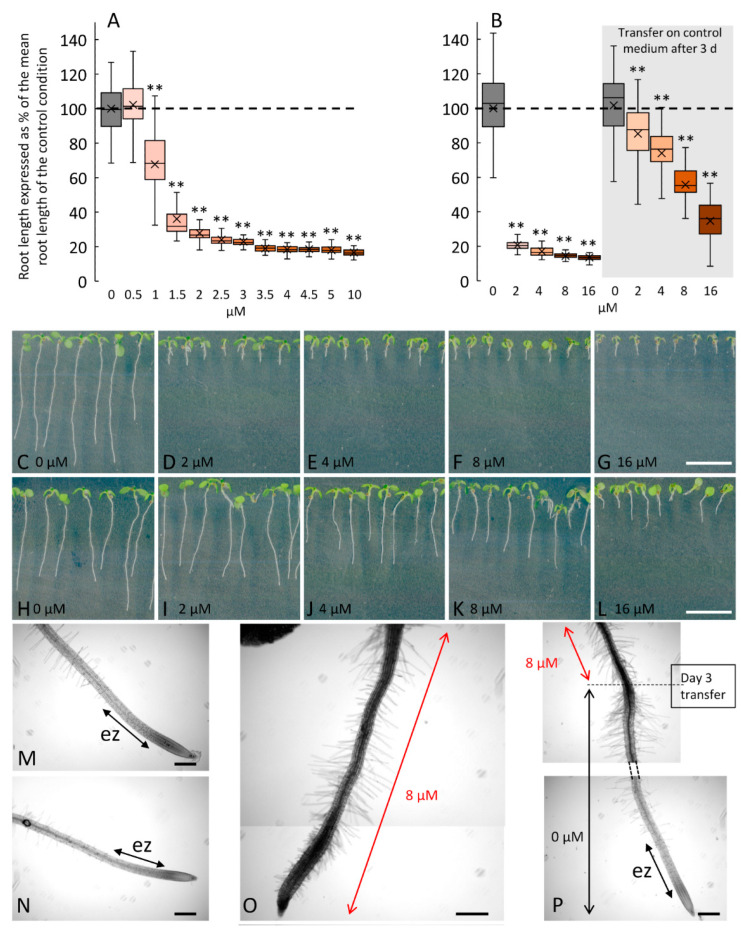
Dose response and reversible effects of 2F-Fuc on *Arabidopsis* root growth after 7 days on treated medium or cultivated for 3 days on treated medium and then transferred on the control medium. (**A**): Changes in root length of 7-day-old *Arabidopsis* cultivated in a medium supplemented with 0 to 10 µM of 2F-Fuc. (**B**): Effect of 2F-Fuc on the root length after 7 days on the treated medium (left panel) or after 3 days on the treated medium and 4 days on the control medium (right panel). (**C**–**I**): Representative pictures of *Arabidopsis* seedlings after 7 days on 0 to 16 µM of 2F-Fuc (**C**–**G**) or after 3 days on 0 to 16 µM of 2F-Fuc followed by 4 days on the control medium (**H**–**L**). (**M**–**P**): Close-up pictures of 7-day-old roots in the control condition (**M**,**N**) or treated with 8 µM of 2F-Fuc (**O**,**P**) without or with transfer at day 3 on the control medium. Scale bars: (**C**–**L**): 10 mm; (**M**–**O**): 200 µm. ez, elongation zone. Dashed line in (**A**,**B**) represents the mean root length in control conditions. ** indicates statistically significant differences from the control condition (*p* < 0.01). (**O**) is a reconstructed image made from two overlapping pictures.

**Figure 4 ijms-24-02533-f004:**
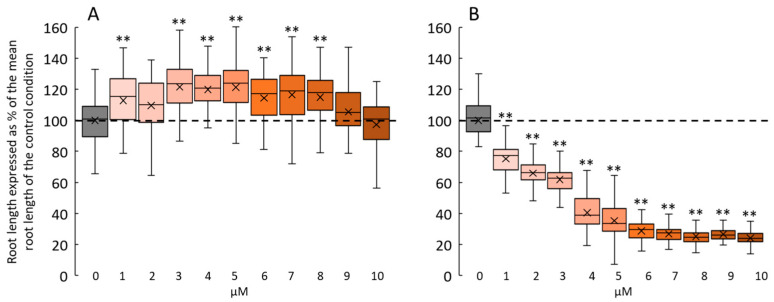
Dose response of 2F-Fuc-NB and photoreleased 1-OH-2F-Fuc on *Arabidopsis* root growth after 7 days. (**A**): Changes in root length in 7-day-old *Arabidopsis* cultivated in a medium supplemented with 0 to 10 µM of 2F-Fuc-NB without photolysis. (**B**): Effect of photoreleased 1-OH-2F-Fuc from 2F-Fuc-NB on 7-day-old roots after 10 min of irradiation at 365 nm at day 3. Dashed lines in (**A**,**B**) represent the mean root length in the control conditions. ** indicates statistical differences from the control condition (*p* < 0.01).

**Figure 5 ijms-24-02533-f005:**
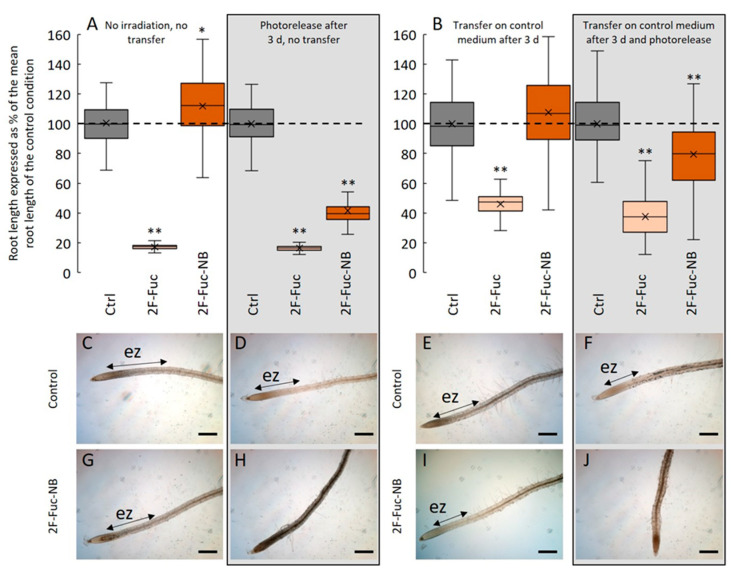
2-F-Fuc-NB is incorporated in root cells before photolysis. (**A**): Effect of 5 µM of 2F-Fuc or 2F-Fuc-NB on the root length with (boxed) or without photolysis. (**B**): Effect of 5 µM 2F-Fuc or 2F-Fuc-NB on the root length after transfer on the control medium followed by photolysis (boxed) or not. **C**–**J**: Representative pictures of *Arabidopsis* roots after 7 days in control conditions (**C**,**D**) or in a medium supplemented with 5 µM of 2F-Fuc-NB (**G**,**H**) without (**C**,**G**) or with (**D**,**H**) photolysis after 3 days. (**E**,**F**): Representative pictures of *Arabidopsis* roots after 3 days of growth followed by a transfer on a new control medium and 4 more days of growth in the control condition with (**F**) or without (**E**) photolysis after the transfer. (**I**,**J**): Representative pictures of *Arabidopsis* roots after 3 days of growth on a medium supplemented with 5 µM 2F-Fuc-NB followed by a transfer on the control medium and 4 more days of growth in the control condition with (**J**) or without (**I**) photolysis after the transfer. Scale bars: (**C**–**J**): 200 µm. ez: elongation zone. Dashed line in (**A**,**B**) represents the mean root length in control conditions. ** indicates statistically significant differencesfrom the control condition (*p* < 0.01).

## Data Availability

Not applicable.

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
