# Peer review of "The Light-Controlled Release of 2-fluoro-l-fucose, an Inhibitor of the Root Cell Elongation, from a nitrobenzyl-caged Derivative"

_ijms, 2023, doi:10.3390/ijms24032533_

Round 1
Reviewer 1 Report
This Paper by Calier et al. constitutes a nice extension of caged sugar analogues for the photoregulation of plants roots grow. Overall, this is an interesting manuscript that could lead to a new strategy for the study of the function of cell wall polysaccharides during cell growth. Therefore, this manuscript could be well suited for a future publication in International Journal of Molecular Sciences. However, I have a few suggestions for improvement that I would like to see in the published version, see my comments below:
It's really difficult to determine the real novelty in the field of caged sugar analogues since the photochemical characterization of this compound is missing (eg. uncaging quantum yield, and yield of sugar liberation). For the yield of 2-fluoro-L-fucose, the 19F NMR analysis is not very informative, I would recommend to add a precise quantification of the photolytic reaction by UV-Visible analysis together with a 1H NMR analysis to fully clarify this point.
I was also surprised to see that without any mechanistic study the authors report a radical pathway for the generation of the NB aci-nitro intermediate. It’s has been well documented that depending on the nature of the leaving group, the NB substituent or the solvent; the aci-nitro intermediate can be formed weather by an [1,5] excited-state intramolecular hydrogen transfer form the triplet or the singlet excitated state of NB (see for example: Helmut Görner [DOI: 10.1111/j.1751-1097.2007.00215.x] or Thomas Bally [DOI: 10.1002/chem.201303338] literature). I understand that a mechanistic study is not the main purpose of this paper, however I recommend to correct the manuscript accordingly.
The selected UV excitation wavelength (UVA) is usually described as detrimental for applications on cells. In this study it’s not clear how this type of irradiation influences the root length since all data were normalized. Therefore, I suggest to clearly comment on the influence of the UVA excitation on the root length.
In addition, the authors claim that the caged compound is efficiently incorporated in the roots, however this is based on a small effect on the root growth (only 20%), I would suggest to use a local irradiation to demonstrate this point nicely. The authors should also precisely describe how many experiments were performed on cells on each figure captions.
Reviewer 2 Report
The main aim of the paper is to study the effect of a modified monosaccharides in root cell elongation metabolism.
The study shows an interesting approach and a simple mechanism to study root cell elongation metabolism. To improve the paper a further discussion of the possible applications can be added. Can it be used in other species? Or just in parts of the plant? Which metabolic processes are inhibited?
Specific comments
Line 44. Figure 1 should be next to the text were is referenced first.
Line 62. Reference needed
Line 88. Specific wavelength compounds can be used to avoid this problem? As it is described in material and methods.
Line 103. Figure 2 should be next to the text were is referenced first.
Line 131. Which are the main differences in the conditions of your study and the other?
Line 228. ** indicates statistical differences with the control plants, but not from different doses?
Round 2
Reviewer 1 Report
The Authors reply to my remarks, and I think the manuscript can now be accepted for publication.